Journal of Data-centric Machine Learning Research (2024)          Submitted 3/24; Revised 06/24; Published 07/24

# *GlycoNMR*: Dataset and Benchmark of Carbohydrate-Specific NMR Chemical Shift for Machine Learning Research

**Zizhang Chen**[1,*]                                                        ZIZHANG2@BRANDEIS.EDU
**Ryan Paul Badman**[2,3,*]                                      RYAN_BADMAN@HMS.HARVARD.EDU
**Lachele Foley**[4]                                                            LFOLEY@UGA.EDU
**Robert Woods**[4]                                                    RWOODS@CCRC.UGA.EDU
**Pengyu Hong**[1]                                                     HONGPENG@BRANDEIS.EDU

[1]*Department of Computer Science, Brandeis Univeristy*

[2]*Department of Neurobiology, Harvard Medical School*

[3]*Kempner Institute, Harvard University*

[4]*Complex Carbohydrate Research Center, University of Georgia*

**Reviewed on OpenReview:** *https://openreview.net/forum?id=N67Uf1b4hm*

**Editor:** Yue Zhao

## Abstract

Molecular representation learning (MRL) is a powerful contribution by machine learning to chemistry as it converts molecules into numerical representations, which is fundamental for diverse downstream applications, such as property prediction and drug design. While MRL has had great success with proteins and general biomolecules, it has yet to be explored for carbohydrates in the growing fields of glycoscience and glycomaterials (the study and design of carbohydrates). This under-exploration can be primarily attributed to the limited availability of comprehensive and well-curated carbohydrate-specific datasets and a lack of machine learning (ML) techniques tailored to meet the unique problems presented by carbohydrate data. Interpreting and annotating carbohydrate data is generally more complicated than protein data and requires substantial domain knowledge. In addition, existing MRL methods were predominantly optimized for proteins and small biomolecules and may not be effective for carbohydrate applications without special modifications. To address this challenge, accelerate progress in glycoscience and glycomaterials, and enrich the data resources of the ML community, we introduce GlycoNMR. GlycoNMR contains two laboriously curated datasets with 2,609 carbohydrate structures and 211,543 annotated nuclear magnetic resonance (NMR) atomic-level chemical shifts that can be used to train ML models for precise atomic-level prediction. NMR data is one of the most appealing starting points for developing ML techniques to facilitate glycoscience and glycomaterials research, as NMR is the preeminent technique in carbohydrate structure research, and biomolecule structure is among the foremost predictors of functions and properties. We tailored a set of carbohydrate-specific features and adapted existing 3D-based graph neural networks to tackle the problem of predicting NMR shifts effectively. For illustration, we benchmark these modified MRL models on GlycoNMR.

---

*. These authors contributed equally to this work.

**Keywords:** AI for science, Glycoscience, Dataset and Benchmark, Graph Neural Networks, and Nuclear Magnetic Resonance.

## 1 Introduction

Considerable efforts have been devoted to developing ML techniques for learning representations of biomolecular structures (Wu et al., 2018; Rong et al., 2020; Méndez-Lucio et al., 2021; Wengert et al., 2021; Yan et al., 2022; Zhou et al., 2023; Guo et al., 2023). Yet, most attention has been devoted to proteins and small biomolecules, while limited progress has been made on carbohydrates despite being the most abundant biomaterials on earth (Oldenkamp et al., 2019). There has been a recent acceleration of interest and progress in carbohydrates in various fields, with findings emphasizing the role of carbohydrate structures in a list of essential medical and scientific topics. Such topics include biological processes of cells (Apweiler et al., 1999; Hart and Copeland, 2010; Varki, 2017), cancer research and treatment targets (Paszek et al., 2014; Tondepu and Karumbaiah, 2022), novel glycomaterials development (Coullerez et al., 2006; Reichardt et al., 2013; Huang et al., 2017; Pignatelli et al., 2020; Richards and Gibson, 2021; Cao et al., 2022) and carbon sequestration in the context of climate change (Pakulski and Benner, 1994; Gullström et al., 2018).

Similar to other biomolecules, the functions and properties of carbohydrates highly depend on their structures. Nevertheless, structure-function relationships remain relatively less understood in carbohydrates than other classes of biomolecules, partly stemming from the bottlenecks in theory and limited structural data (Ratner et al., 2004; Hart and Copeland, 2010; Oldenkamp et al., 2019). In chemical sciences, nuclear magnetic resonance (NMR) is the primary characterization technique used for determining atomic-level fine structures of carbohydrates. It requires correctly interpreting solution-state NMR parameters, such as chemical shifts and scalar coupling constants (Duus et al., 2000; Brown et al., 2018). NMR still relies on highly trained personnel and domain knowledge, due to limitations in theoretical understanding (Lundborg and Widmalm, 2011; Toukach and Ananikov, 2013). This opens up an opportunity for ML research. ML methods are relatively under-explored in carbohydrate-specific studies and especially for predicting the NMR chemical shifts of carbohydrates (Cobas, 2020; Jonas et al., 2022). Improving the efficiency, flexibility, and accuracy of ML tools that relate carbohydrate structures to NMR parameters is well-aligned to recently launched research initiatives such as GlycoMIP (`https://glycomip.org`), a National Science Foundation Materials Innovation Platform that promotes research into glycomaterials and glycoscience, as well as parallel efforts by the European Glycoscience Community (`https://euroglyco.com`).

ML methods have significant potential for generality, robustness, and high-throughput analysis of biomolecules, as demonstrated by a plethora of previous works (David et al., 2020; Shi et al., 2021; Jonas et al., 2022). Inspired by the recent successes of MRL in various fields and applications, such as molecular property prediction (Rong et al., 2020; Yang et al., 2021a; Zhang et al., 2021), molecular generation (Shi et al., 2020; Zhu et al., 2022; Zhou et al., 2023), and drug-drug interaction (Chen et al., 2019; Lyu et al., 2021), we embarked on the journey toward building ML tools for predicting carbohydrate NMR chemical shift spectra from the perspective of molecular representation learning. The first and foremost technical barrier we encountered was the issues with the quality, size, and

accessibility of carbohydrate NMR datasets–ongoing problems which have been pointed out in recent literature (Toukach and Ananikov, 2013; Toukach and Egorova, 2019; Ranzinger et al., 2015; Toukach and Egorova, 2022; Böhm et al., 2019). Much of the current data limitations result from the fact that carbohydrates are the most diverse and complex class of biomolecules. Their numerous chemical properties and configurations make the annotation and analyses of their NMR data substantially more complicated and uncertain than those for other biomolecules (Herget et al., 2009; Hart and Copeland, 2010). Particularly, existing structure-related carbohydrate NMR spectra databases are less extensive and less accessible to ML researchers than databases for other classes of biomolecules and proteins, leading to recent calls for improvement in standards and quality (Ranzinger et al., 2015; Paruzzo et al., 2018; Böhm et al., 2019; Toukach and Egorova, 2022). Among all NMR signals, atomic 1D chemical shifts are the most accessible and generally complete (Jonas et al., 2022) and provide rich information to enable molecular identification and fingerprint extraction for carbohydrates.

To facilitate an initial convergence of ML, glycoscience, and glycomaterials, we have developed `GlycoNMR`, a data repository of carbohydrate structures with curated 1D NMR atomic-level chemical shifts. `GlycoNMR` includes two datasets. In the first one, we manually curated the experimental NMR data of carbohydrates available at `Glycosciences.DB` (formerly SweetDB) (Loß et al., 2002; Böhm et al., 2019). The second one was constructed by processing a large sample of NMR chemical shifts we simulated using the Glycan Optimized Dual Empirical Spectrum Simulation (GODESS) platform (Kapaev and Toukach, 2015, 2018), which was partly built on the Carbohydrate Structure Database (CSDB) (Toukach and Egorova, 2019, 2022). Substantial domain expertise and efforts were involved in both annotating and preprocessing the two datasets. To the best of our knowledge, `GlycoNMR` is the first large, high-quality carbohydrate NMR dataset specifically curated for ML research. Using `GlycoNMR`, we designed a set of features, particularly for describing the structural dynamics of carbohydrates, and developed a baseline 2D GNN model for predicting carbohydrates' 1D NMR chemical shifts. In addition, we adapted five state-of-the-art 3D-based MRL models to align with the atomic-level NMR shift prediction and benchmarked their performances on `GlycoNMR`. The experimental results demonstrate the feasibility and promise of ML in analyzing carbohydrate NMR data, and, more generally, in advancing the development of glycoscience and glycomaterials.

**Summary of contributions:**

- We developed `GlycoNMR`, a large, high-quality, ML-friendly carbohydrate NMR dataset that is publicly and freely available at https://github.com/Cyrus9721/GlycoNMR. Instructions for loading and fitting data to GNN models are in Appendix N.

- We designed a set of chemically-informed features that are tailored specifically for carbohydrates. In addition, we experimentally showed that these features can intrinsically capture the unique structure dynamics of carbohydrates and thus enhance the performance of graph-based MRL models.

- We adapted and benchmarked multiple 3D-based MRL methods on `GlycoNMR` and demonstrated the potential usage of ML approaches in glycoscience research. Demos are provided in Appendix O.

## 2 Background and Related Work

**Carbohydrates:** Carbohydrates (examples in Figure 1), also called saccharides, are one of the major biomolecule classes aside from proteins, lipids, and nucleic acids on earth. At the macroscale, carbohydrates are common in sugars and digestive fibers in our diets, and at the microscale, they are widespread on cell membranes and in metabolic pathways. Monosaccharides (introduced in Figure 1), also known as simple sugars (Chaplin, 1986), are the base units of carbohydrates and are typically composed of carbon, hydrogen, and oxygen atoms in specific ratios. Glycosidic bonds, which link monosaccharides into chains or trees, are formed via condensation reactions between the connected monosaccharides. Long chains of monosaccharides are also called polysaccharides. Structure patterns closely relate to the NMR chemical shift spectra and functions (Blanco and Blanco, 2022) of carbohydrates. Importantly, five attributes are the minimum structural information necessary to describe a monosaccharide in a given carbohydrate: (1) Fischer configuration, (2) stem type, (3) ring size, (4) anomeric state, and (5) type and location of modifications (Herget et al., 2009), additional features may be helpful, as discussed in Table 2 and Appendix B. Furthermore, the central ring carbon atoms and their corresponding hydrogen atoms are labeled in a universal and formulaic way in carbohydrates, which aids in building carbohydrate-specific features in the ML pipeline.

**Formula**: aLFucp(1-3) [bDGalp(1-4),Ac(1-2)] bDGlcpN(1-3) bDGalp(1-4) aDGlcp

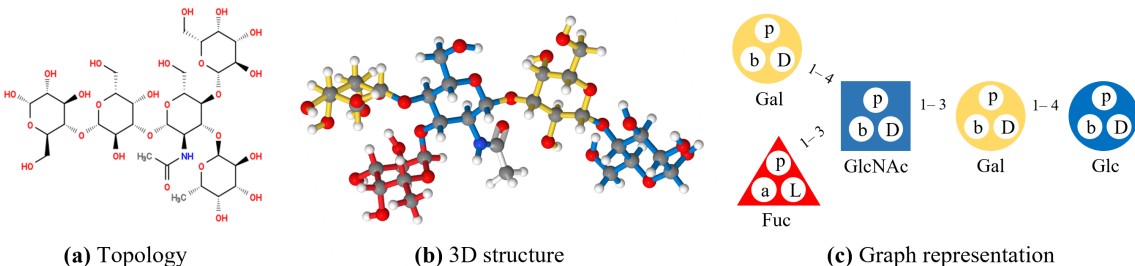

**(a)** Topology        **(b)** 3D structure        **(c)** Graph representation

Figure 1: An example carbohydrate containing 5 monosaccharides (formula in the top): **(a)** The topology; **(b)** The 3D structure with nodes and edges indicating atoms (gray: C, red: O, white: H, blue: N) and bonds, respectively; **(c)** The graph representation. The big graph nodes indicate the monosaccharide stems (yellow circle: Gal, blue circle: Glc, blue square: GlcNAc, and red triangle: Fuc) connected by edges labeled with glycosidic linkages ("1-3" or "1-4"). "D"/"L" indicate isomers information, "a"/"b" indicate anomers, and "p" indicates the ring size.

**Nuclear Magnetic Resonance (NMR):** NMR spectra provide key structural features of carbohydrates, including the stereochemistry of monosaccharides, glycosidic linkage types, and conformational preferences. Its non-destructive nature, high sensitivity, and ability to analyze samples in solution make NMR an indispensable tool for carbohydrate research. Arguably, the most accessible and complete NMR parameter for computational structural studies is the 1D chemical shifts (Jonas et al., 2022), where in carbohydrates, usually only the hydrogen $^1$H and carbon $^{13}$C nuclei shifts are measurable (Toukach and Ananikov, 2013). Figure 2 shows a simple carbohydrate and its $^1$H and $^{13}$C NMR spectra. As another challenge

specific to carbohydrates, carbohydrate NMR peaks are constrained to a much narrower region of spectra range than proteins, making them harder to separate and leading to an over-reliance on manual interpretation (Toukach and Ananikov, 2013). Thus, the development of theoretical and computational tools that can more automatically and accurately relate a carbohydrate structure and its NMR parameters is a high priority for the field (Hart and Copeland, 2010; Herget et al., 2009; Toukach and Ananikov, 2013; Jonas et al., 2022).

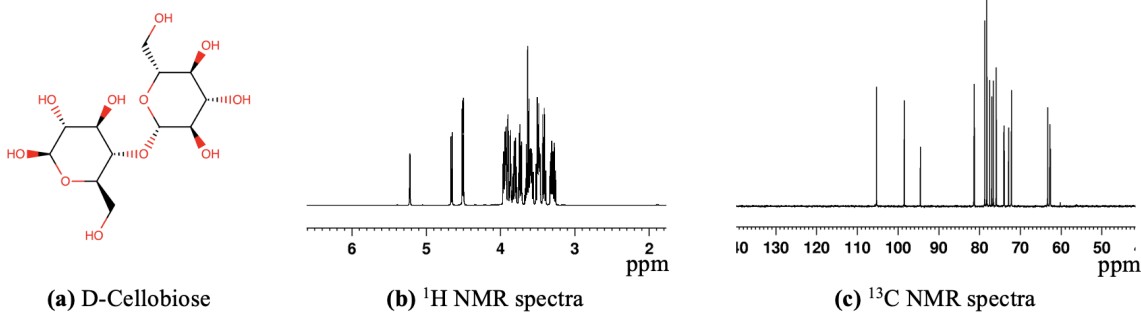

**(a)** D-Cellobiose       **(b)** [1]H NMR spectra       **(c)** [13]C NMR spectra

Figure 2: D-Cellobiose (**a**) and its NMR spectra (**b**) & (**c**). D-Cellobiose comprises two glucose units in beta (1-4) glycosidic linkage and is a natural product found in *Aspergillus genus*.

**Structure-related Chemical Shift Prediction Methods:** The primary computational methods for chemical shift prediction of carbohydrates can be grouped into four categories: ab initio methods, rules-based and additive increment-based methods, substructure codes, and data-driven ML approaches (Jonas et al., 2022). Ab initio methods, such as density functional theory (DFT) methods, are so far the most accurate as they are based on foundational physics and chemistry theory, but have the lowest throughput and often require considerable expert parameter sweeping and tuning (Tantillo, 2018; Kevin et al., 2019). Additive increment-based approaches, such as CASPER (Jansson et al., 2006; Lundborg and Widmalm, 2011), rely on carefully designed rules, which have limited generalization power and have yet to be extensively validated against experimental data (Jonas et al., 2022). Substructure codes, such as HOSE codes (Kuhn et al., 2008), are the oldest prediction method and still can provide competitive performance in some cases (Jonas et al., 2022). However, substructure code methods have limitations in encoding stereochemical information and distinguishing conformers (though improvements were made recently in this area (Kuhn and Johnson, 2019)). Most HOSE code methods are based on neighborhood search that requires closely similar examples to reach adequate prediction quality (Kuhn et al., 2008). HOSE codes were tested with several large experimental datasets containing assorted biomolecules, with accuracy ranging from approximately 1-3.5 ppm for [13]C and 0.15-0.30 ppm for [1]H (mean absolute error) (Jonas et al., 2022) (performance of NMR chemical shift prediction on carbohydrates has not been reported to our knowledge). GODESS is a high-quality carbohydrate-specific hybrid of HOSE-like methods and rules-based methods for NMR prediction (Kapaev and Toukach, 2015, 2018). GODESS is capable of generating both structure files in standard carbohydrate format, and atomic-level NMR chemical shift predictions for central ring carbon and hydrogen atoms as well as for some modification group atoms. In this study, we used GODESS to produce the experimentally informed simulated data in GlycoNMR.Sim.

Lastly, ML methods, especially graph neural networks (GNNs) (Battaglia et al., 2018; Zhou et al., 2020; Zhang et al., 2023), have shown great potential for predicting NMR spectra for biomolecules (Jonas and Kuhn, 2019; Kang et al., 2020; Yang et al., 2021b; McGill et al., 2021; Jonas et al., 2022). Nevertheless, they are relatively unexplored in carbohydrates. To fill in this gap, this paper presents, to our knowledge, the first ML-based attempt tailored specifically to predict NMR chemical shifts for carbohydrates.

**Relation to ML in other biomolecules:** A large range of papers exists for tackling problems in NMR with ML broadly. Comprehensive general reviews of ML applications in NMR in recent years include (Chen et al., 2020; Bratholm et al., 2021; Yokoyama et al., 2022; Kuhn, 2022; Li et al., 2022; Cortés et al., 2023). The most extensive review of ML to predict NMR spectra of biomolecules is (Jonas et al., 2022) (especially Table 1 in the review). CNNs, MPNNs, and $\delta$ machine were found to have the best performance as a general recent trend in NMR for diverse biomolecules (Jonas and Kuhn, 2019; Dračínský et al., 2019; Kwon et al., 2020; Li et al., 2021), though large differences in sample size and dataset composition make firm conclusions hard to draw (low statistics plague the NMR field due to issues in public datasets). IMPRESSION (Gerrard et al., 2020) and CASCADE (Guan et al., 2021) are also notable recent achievements in this area, though their performance for larger molecules remains unknown. GNNs are by far the most commonly recently used ML tools in this area, though, while feedforward networks dominated earlier work (Jonas et al., 2022).

**Relation to Graph-based MRL:** Graph-based MRL has gained accelerating amounts of attention due to its ability to capture local connectivity and topological information of biomolecules (Gilmer et al., 2017; Kipf and Welling, 2017; Hamilton et al., 2017; Xu et al., 2019; Veličković et al., 2018). In graph-based MRL, molecules can be encoded in either 2D or 3D graphs with atoms as nodes. In a 2D molecular graph, edges can be pre-determined chemical bonds. In a 3D molecular graph, edges are determined based on the 3D coordinates of atoms, to capture their atomic interactions. Several message-passing schemes have been developed for GNNs to use spatial information such as atom interactions (Schütt et al., 2017), bond rotations and angles between bonds (Gasteiger et al., 2020b,a; Wang et al., 2022), spherical coordinate systems (Liu et al., 2022) and topological geometries (Fang et al., 2022). We chose GNN-based MRL models as the baseline due to their strong expressive power and promising performance on other kinds of biomolecules. For example, encoding bond distance as continuous numbers is expensive computationally, so GNNs encode simplified bond information to avoid the full computational cost (Yang et al., 2021b). Chemists also know that atoms in carbohydrates interact non-negligibly up to 3-4 atoms away, and GNN node-edge structures are well-posed to account for these interactions (Kapaev and Toukach, 2015). We thus formulated the structure-related chemical shift prediction problem as a node regression task, and trained GNNs to predict the chemical shifts of primary monosaccharide ring carbons and hydrogen atoms. We used the Root-Mean-Square Error (RMSE) to evaluate the predicted and the ground truth chemical shifts.

## 3  Datasets and Baseline Model

We designed a pipeline specifically for annotating the NMR data that we gathered to facilitate the development of ML techniques for predicting atomic NMR shifts of carbohydrates. Two

ML-friendly NMR chemical shift datasets **GlycoNMR.Exp** and **GlycoNMR.Sim** were constructed. They contain both the 3D structures and the curated $^1$H and $^{13}$C NMR chemical shifts of the carbohydrates. The data statistics are summarized in Table 1. In addition, Table 5 and Figure 5 (in Appendix) show the list of covered monosaccharides and a histogram of carbohydrate sizes (number of monosaccharides) in our datasets. A set of carbohydrate-specific features was engineered to describe the atom structure dynamics in a carbohydrate. We incorporated the features to build a baseline model (see Section 3.2).

**GlycoNMR.Exp:**   This dataset mainly contains the experimental NMR data obtained from Glycosciences.DB (Böhm et al., 2019). Glycosciences.DB inherited data from the discontinued Complex Carbohydrate Structure Database (CCSD/CarbBank) (Doubet and Albersheim, 1992). It was semi-automatically populated (with moderator oversight) by the carbohydrate entries in the worldwide Protein Data Bank (wwPDB) (Böhm et al., 2019). The NMR data was supplied by SugaBase (Vliegenthart et al., 1992) or manually uploaded by researchers. Glycosciences.DB contains around 3400 carbohydrate entries associated with NMR shifts (however, most only with partial annotation on $^1$H or $^{13}$C). We found 299 carbohydrates had both structures and complete (or near complete) $^1$H and $^{13}$C shifts, and included them in GlycoNMR.Exp for further annotation and processing to make the dataset ML-friendly (details in Section 3.1). This processing required substantial domain expertise and efforts on our part, due to the inconsistent and sometimes ambiguous labeling and organization of the diversely-sourced data from this repository. For better illustration, we present a medium-sized carbohydrate data example, including both its raw data file 1, 2 and annotated and processed file. Notice that raw data file 1, 2 records the carbohydrate structure and NMR shift separately.

**GlycoNMR.Sim:**   This dataset contains the simulated NMR chemical shifts produced by using GODESS (`http://csdb.glycoscience.ru`) (Toukach and Egorova, 2016). GODESS combines incremental rule-based methods (called "empirical" simulation in GODESS) and/or HOSE-like "statistical" methods, and is informed by the CSDB experimental data (Kapaev et al., 2014; Kapaev and Toukach, 2016, 2018; Toukach and Egorova, 2022). GODESS recently demonstrated superior performance in simulating certain carbohydrate NMR shifts and could sometimes perform comparably to DFT (Kapaev et al., 2014; Kapaev and Toukach, 2016, 2018). Hence, we chose it to produce a simulation dataset to amend the lack of publicly available experimental NMR data for carbohydrates. GODESS requires the formula of carbohydrates to be written in the correct CSDB format (Toukach and Egorova, 2019), and does not produce results for those formulas that it deems chemically impossible or incorrect. Helpfully, GODESS scores the trustworthiness of each simulation result. We excluded simulation results with low trustworthiness (error > 2 ppm). We were able to simulate and curate NMR chemical shifts for ∼200,000 atoms in 2,310 carbohydrates. For a demonstration, we present a large-sized carbohydrate, including both its simulated raw data files 1, 2, 3 and the annotated and processed file. Raw data file 1 contains the structural information of the carbohydrate, while the $^{13}$C and $^1$H NMR chemical shifts are stored separately in the raw data files 2 and 3, respectively. In Appendix I, we evaluate the quality of the GODESS simulation by comparing the simulated and experimental data of carbohydrates shared by GlycoNMR.Exp and GlycoNMR.Sim. The results indicate that GlycoNMR.Sim is valuable to a certain extent.

The GlycoNMR datasets are much larger than those used in most carbohydrate-specific studies, which typically have $< 100$ molecules (Furevi et al., 2022). The size of our data is also comparable to those of the protein or biomolecule NMR datasets, which usually number in the hundreds to low thousands of molecules at best (see Table 1 in (Jonas et al., 2022), or (Yang et al., 2021b)).

Table 1: Dataset Statistics. In total, GlycoNMR contains 2,609 carbohydrate structures with 211,543 atomic NMR chemical shifts. The average molecular size of GlycoNMR.Exp is 91.2 and of GlycoNMR.Sim is 161.5, which is much larger than those of molecules that are commonly used in MRL (Wu et al., 2018). In publicly available data, central ring carbons are the most consistently reported. Hence, we focused on those central ring carbons and the attached hydrogen atoms in this study. Additionally, we observed that the central ring atom-level shift values are often missing in the NMR data files obtained from Glycosciences.DB as the original experimental data was sometimes not completely interpreted, which is an ongoing issue in carbohydrate research using NMR. In our Glycosciences.DB subset only 3% of all central ring atoms were blanked out due to missing chemical shifts.

| Data source | # Carbohydrate | # Monosaccharide | # Atom | # labeled NMR shifts |
|---|---|---|---|---|
| *GlycoNMR.Exp* | 299 | 1,130 | 27,267 | 11,848 |
| *GlycoNMR.Sim* | 2,310 | 16,030 | 372,958 | 199,695 |

## 3.1 Data Annotation

We performed extensive data annotation to associate the 3D structure of each carbohydrate with its NMR chemical shifts. GlycoNMR.Exp and GlycoNMR.Sim store the structure data for each carbohydrate in the Protein Data Bank (PDB) format amended for carbohydrates, which contains comprehensive 3D structural information, including atom types, ring positions, 3D atomic coordinates, connectivity of the atom, and three-letter abbreviations for monosaccharides. The corresponding NMR data for a given carbohydrate is stored in a separate file, which contains: (1) the hydrogen and carbohydrate chemical shifts per atom per monosaccharide unit and (2) the lineage information of each monosaccharide to its root. We first matched the monosaccharides in the PDB file with those in the NMR file and then used the ring position information to match atoms across the PDB and NMR files. Unfortunately, the order of the monosaccharides in the PDB file often does not match that in the NMR file. The three-letter abbreviations of monosaccharides in PDB files increase the matching difficulty due to the inherent ambiguity and inconsistency in carbohydrate naming convention across time and labs (Toukach and Egorova, 2019). In addition, one carbohydrate can contain multiple monosaccharide units of the same type. For example, in the Glycosciences.DB-sourced PDB files, a single type of monosaccharide can manifest as multiple residues with identical three-letter coding names or vice versa. Hence, we had to utilize domain knowledge to reduce such ambiguities as much as possible when handling the Glycosciences.DB data. Furthermore, we validated topological connections between monosaccharides using the PDB structure data (see Figure 3). We matched the topological connections between monosaccharide units in the PDB files from Glycosciences.DB and GODESS with the lineage information used by their corresponding NMR data files. This

allowed us to match the monosaccharides in the PDB files with those in the NMR files, and then use ring positions to associate atoms in a PDB file with atoms in the corresponding NMR data file. Detailed annotation process recorded in Appendix N, including GitHub repos and examples for data annotation.

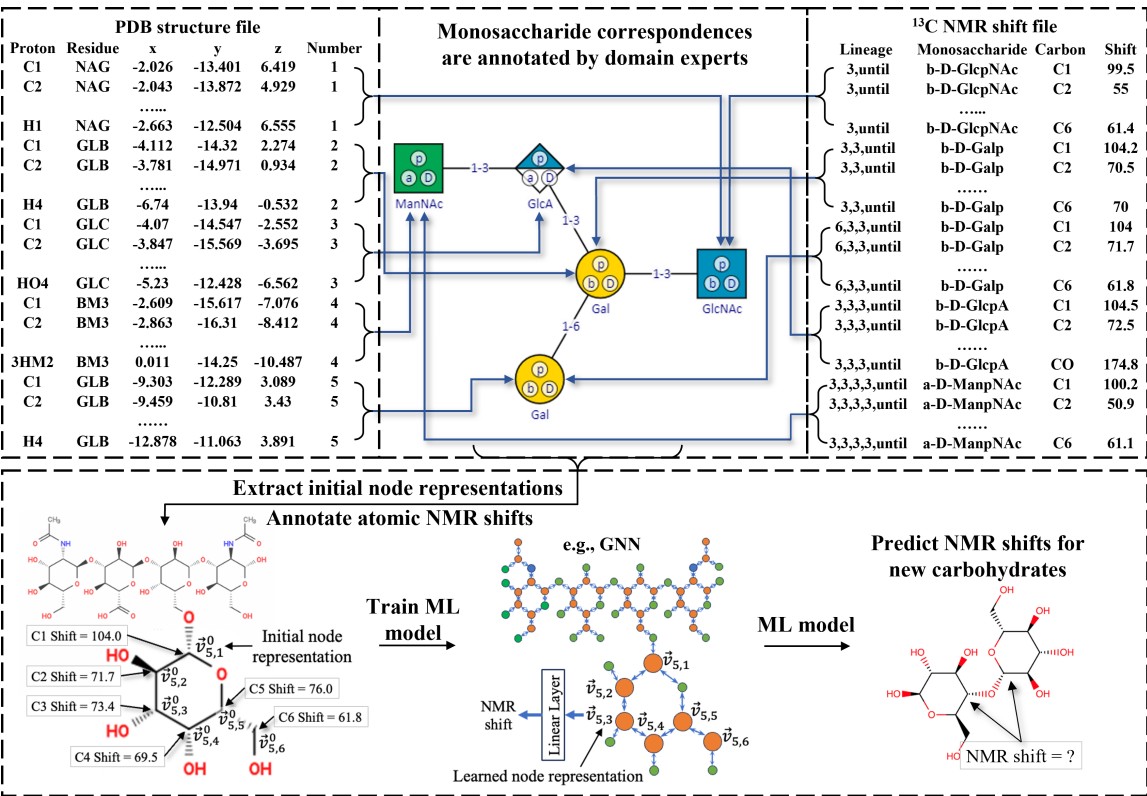

Figure 3: A key task in data annotation is matching monosaccharides in each carbohydrate's PDB and NMR files. Conceptually, matching is done by linking monosaccharides in the PDB file (**top left**) and the NMR file (**top right**) to their topological positions in the carbohydrate (**top middle**). The "Residue" column in the PDB file contains the 3-letter abbreviations of monosaccharides. Once this matching is established, we can use atom types and their ring positions to assign the chemical shifts in the NMR file to their atoms in the PDB file (**bottom left**). All features are encoded at the atom level (**bottom middle**), and the model predicts the atomic shifts (**bottom right**).

## 3.2 Glycoscience-informed Feature Engineering

We derived a set of structural features for the atoms in carbohydrates, which are categorized into monosaccharide-level and atom-level (see Table 2). The monosaccharide-level features describe the monosaccharide context of an atom, including the stem type, configuration, ring size, anomeric status of the monosaccharide, and modifications to the monosaccharide. These

Table 2: Features designed for carbohydrates.

| Feature | Explanation | Example values |
|---|---|---|
| **Monosaccharide** | | |
| Configuration | Fischer convention | D, L |
| Stem type | Basic monosaccharide unit | Gal, Glc, Man, .. |
| Ring size | Number of ring carbons | Pyranose ($p$), Furanose ($f$) |
| Anomer | Anomeric orientation hydroxyl group | $\alpha, \beta$ |
| Modifications | Modification groups | Ac, Sulfate, Me, Deoxygenation |
| **Atom** | | |
| Ring position | Atom position in carbon ring | C1, C2, C3, C4, ... |
| Atom type | Chemical elements | C, H, N, O, ... |

features encode the stereochemistry properties and structural dynamics of a monosaccharide and provide information about the overall electronic environment of each atom. The atom-level features include the ring position (wherein the ring the atom is located or is attached) and atom type. Ring position indices (e.g. C1, C2, etc.) are standardized and consistent in carbohydrate structure labeling. To briefly introduce the ring position of an atom, monosaccharide units are classified as either aldoses or ketoses. The aldehyde carbon in aldoses is always numbered as C1 and the ketone carbon in ketoses is labeled as the lowest possible number (Fontana and Widmalm, 2023). The ring position provides information about what other atoms and/or functional groups the atom interacts with. Both categories of features play a significant role in determining the NMR chemical shift value of the atom. We enhanced the carbohydrate PDB structure files by adding the above features and subsequently converted these enriched files into a tabular format where each row describes an atom along with its 3D coordinate and its features and the features associated with the monosaccharide it belongs to. Table 6 in the Appendix describes the processed PDB file to illustrate the above feature engineering effort.

### 3.3 Evaluation metric

To evaluate the performance of several MRL models in NMR shift prediction, we calculate the Root-Mean-Square Error (RMSE) between the predicted NMR chemical shifts and the ground truth NMR shifts. The formula for calculating the RMSE is provided in Appendix G. In addition, RMSE is sensitive to outliers, and it can also help us find mismatches caused by humans, especially for the experimental dataset that requires extensive data annotation. For example, we repeatedly applied an outlier check by comparing the ground truth NMR shift and the predicted NMR shift generated by the baseline model to develop our preprocessing pipeline.

### 3.4 Baseline Model on 2D Graph Neural Network

We adopted a 2D graph convolutional neural network (Kipf and Welling, 2017) as our baseline model, a standard architecture that is easy to build off of. We added two linear

layers (one for input and the other for output). Each carbohydrate is represented as a graph, with nodes representing atoms and edges representing bonds between atoms. Each node is associated with the features described above. If a carbohydrate structure file does not provide information about the bond connectivity between atoms, we added edges between atom nodes based on their distances. The distance thresholds are 1.65Å for C-C (Liu et al., 2021), 1.18Å for H-X (Guzmán-Afonso et al., 2019), and 1.5Å for X-X (Gunbas et al., 2012), where X stands for other atoms. For each of the GlycoNMR.Exp and GlycoNMR.Sim datasets, we randomly split the carbohydrates into the 80/20 train/validation subsets. Using the training subset, a model was trained to predict $^{13}$C or $^1$H NMR chemical shifts and was evaluated on the corresponding validation subset. The validation results are presented in Figure 4, showing that the baseline model performs reasonably well despite its relatively simple model architecture.

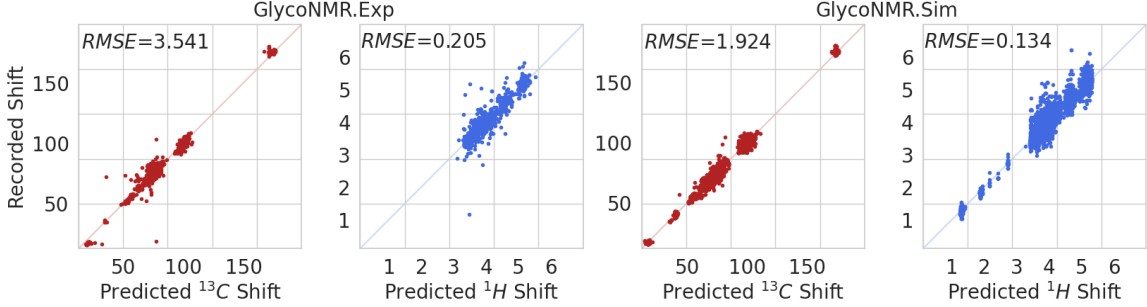

Figure 4: Validation results of the 2D baseline model. In all plots, the horizontal and vertical axes indicate the predicted atomic NMR chemical shifts and the ground truth, respectively (in chemical shift units of ppm). Each point represents an in-ring atom of carbohydrates from the test subset.

We investigated the impact of individual features (described in Section 3.2) on the baseline model by examining the change in the model performance after removing one single feature. We reported the results in Table 3. We observed that the model performance dropped drastically after removing the ring position feature. Furthermore, when either the stem type feature or the anomer feature is removed, the model performance on carbon drops by a relatively large margin. Other features also had impacts, although their effects are relatively mild. This observation resonates with our expectation when designing the features that the ring position should encode important information about the local context of an atom. Note that removing the 3D-related feature "configuration" slightly improves the performance of the 2D model. The results here provide a direction for improvements, for example, designing new structural features and increasing the interpretability of the model.

## 4 Benchmark Study of `GlycoNMR` on 3D Graph Neural Networks

To investigate the benefits of encoding 3D structural information, we adapted and benchmarked five state-of-the-art 3D graph-based MRL methods on our datasets: SchNet (Schütt et al., 2017), DimeNet++ (Gasteiger et al., 2020a), ComENet (Wang et al., 2022), SphereNet (Liu et al., 2022), CASCADE (Guan et al., 2021). The first four models were originally designed

Table 3: Ablation study on the carbohydrate-informed features for the baseline model on the GlycoNMR.Sim and GlycoNMR.Exp datasets. Each column reports the RMSE after removing the feature indicated by the column header. Due to atom-to-modification matching ambiguity in the more inconsistent PDB files used for annotating GlycoNMR.Exp, feature 'Modification' is not used.

| GlycoNMR.Exp | Ring position | Modification | Stem type | Anomer | Configuration | Ring size | None |
|---|---|---|---|---|---|---|---|
| $^1$H | 0.376 | N/A | 0.271 | 0.240 | 0.206 | 0.220 | **0.205** |
| $^{13}$C | 20.218 | N/A | 4.475 | 3.749 | 3.461 | 3.575 | **3.541** |

| GlycoNMR.Sim | Ring position | Modification | Stem type | Anomer | Configuration | Ring size | None |
|---|---|---|---|---|---|---|---|
| $^1$H | 0.507 | 0.137 | 0.185 | 0.187 | 0.136 | 0.135 | **0.134** |
| $^{13}$C | 20.5529 | 1.991 | 2.827 | 2.258 | **1.910** | 1.977 | 1.924 |

to predict the graph-level quantum properties of small molecules from their structures. To apply them to our tasks, we replaced their global pooling layer, which is needed for predicting the properties of whole molecules, and added a layer that maps the learned embedding of each atom to its NMR chemical shift. CASCADE is based on SchNet and was developed to use a different feature set. To make it work with our data, we replaced its atom features with ours as described in Section 3.2. For each GlycoNMR dataset, we trained two models for predicting the $^{13}$C and $^1$H NMR shifts, respectively. We randomly partitioned the carbohydrates into an 80/10/10 split for training, validation, and testing. We tuned each model's hyperparameters using the validation set and selected the combination that yielded the best performance. The hyperparameters were selected from the following ranges: learning rate [0.001, 0.01], batch size: [2, 4, 8], number of layers: [2, 3, 4], hidden channel size: [32, 64, 128, 256], and the cut-off distance for deciding the interactions between atoms: [4.0, 5.0, 6.0]. Early stopping based on the performance of the validation subset was used during training. The optimal model was then run five times, and we reported the average RMSE on the test set along with its standard deviation.

We evaluated the models under two settings. In the first one, the representation of each atom is initialized with the atomic-level features as reported in Section 3.2. This adheres to the initialization method outlined in numerous existing publications on MRL, including (Zhou et al., 2023; Liu et al., 2022). In the second setting, we incorporated monosaccharide-level features (see Section 3.2) into the initial atom representations, detailed in Appendix B and further discussed in Table 3. The test results are reported in Table 4 (rows 1-4 show the results of the first setting, and rows 5-8 for the second setting). The running time is compared in Appendix M. We also provided additional multi-task learning results in Appendix L, where we trained one MRL model for each dataset to predict both $^{13}$C and $^1$H shifts.

We observed on GlycoNMR.Sim, under the first setting, DimeNet++ achieved the lowest RMSE on both NMR $^{13}$C and $^1$H NMR shift prediction. In addition, CASCADE obtained the lowest RMSE on the $^{13}$C shift prediction task under the second setting, and DimeNet++ had the lowest RMSE on predicting $^1$H shifts. We noticed a marginal overall improvement in model performance with the incorporation of the monosaccharide-level. On the GlycoNMR.Exp dataset, under the first setting, SchNet performed the best on the $^{13}$C shift prediction task, and CASCADE achieved the lowest RMSE on the $^1$H shift prediction

task. Under the second setting, SphereNet performed the best on predicting $^{13}$C shifts, and CASCADE was the best on predicting $^1$H shifts. A slight drop in performance was observed for some models when the monosaccharide-level features were concatenated into the node-level features. The overall performance of 3D GNN was better than 2D. Thus, we believe that exploring better methods to integrate 3D structural information is a promising direction to enhance NMR chemical shift prediction models.

Table 4:   NMR chemical shift prediction benchmark using 3D MRL methods (in RMSE).

|  | GlycoNMR.Sim | | GlycoNMR.Exp | |
| --- | --- | --- | --- | --- |
|  | $^{13}$C | $^1$H | $^{13}$C | $^1$H |
| ComENet (Wang et al., 2022) + *atom feat.* | 1.4239±0.0229 | 0.1170±0.0005 | 2.7013±0.1424 | 0.1487±0.0093 |
| DimeNet++ (Gasteiger et al., 2020a) + *atom feat.* | **1.3866±0.0092** | **0.1201±0.0005** | 2.6633±0.0638 | 0.1391±0.0034 |
| SchNet (Schütt et al., 2017) + *atom feat.* | 1.4477±0.0079 | 0.1309±0.0017 | **2.3685±0.0267** | 0.1484±0.0020 |
| SphereNet (Liu et al., 2022) + *atom feat.* | 1.4474±0.0015 | 0.1223±0.0015 | 2.5860±0.0993 | 0.1834±0.0332 |
| CASCADE (Guan et al., 2021) + *atom feat.* | 1.4455±0.0089 | 0.1310±0.0006 | 2.3559±0.0673 | **0.1372±0.0024** |
|  |  |  |  |  |
| ComENet (Wang et al., 2022) + *extra feat.* | 1.2913±0.0196 | 0.1058±0.0013 | 2.6250±0.1211 | 0.1458±0.0082 |
| DimeNet++ (Gasteiger et al., 2020a) + *extra feat.* | 1.2379±0.0188 | **0.1042±0.0024** | 3.0073±0.4308 | 0.1489±0.0118 |
| SchNet (Schütt et al., 2017) + *extra feat.* | 1.2212±0.0055 | 0.1187±0.0004 | 2.6590±0.0607 | 0.1630±0.0069 |
| SphereNet (Liu et al., 2022) + *extra feat.* | 1.2362±0.0135 | 0.1087±0.0016 | **2.5860±0.0993** | 0.1834±0.0332 |
| CASCADE (Guan et al., 2021) + *extra feat.* | **1.2075±0.0033** | 0.1195±0.0008 | 2.7392±0.0890 | **0.1381±0.0025** |

In the recent protein and small biomolecule computational studies, the MAEs of NMR chemical shift prediction tasks range from 0.1-0.3 ppm for $^1$H and  0.7-4 ppm for $^{13}$C (Jonas et al., 2022), which depend on specifics of models and datasets (e.g., molecular characteristics, sample size and diversity, simulated or experimental data). SphereNet and CASCADE results from our carbohydrate-specific data are comparable to the reported results on other classes of biomolecules. This computational error range can be compared to an experimental error reported for NMR data collection, which was estimated to be 0.51 ppm for $^{13}$C and 0.09 ppm for $^1$H, according to the mean absolute error across  50,000 shifts in the nmrshiftdb2 (which contains a wide variety of biomolecules) (Jonas and Kuhn, 2019). Although it is difficult to compare errors across studies on different molecule classes, and across laboratory conditions and NMR instruments (Stavarache et al., 2022), these observations suggest that the results reported in Table 4 could serve as a useful reference for future efforts in MRL on carbohydrates.

## 5 Discussion and Conclusions

Carbohydrate research has historically lagged behind other major molecular classes (e.g. proteins, small molecules, DNA, etc.) due to theoretical bottlenecks and data quality issues. To help improve this situation, we introduced the first ML-friendly, carbohydrate-specific NMR dataset (GlycoNMR) and pipelines for encoding carbohydrates in predictively powerful GNNs. We hope this study immediately provides useful resources for ML researchers to engage in this new frontier and form a new force to make glyco-related sciences one of the main applications that drive ML research.

As a limitation of the current dataset, most experimentalists only upload NMR spectra peak positions (which is what we predicted), and raw spectral files are rarely openly provided in carbohydrate research. However, the full peak shapes (e.g., widths, heights) and broader

spectral patterns also encode rich structural information, but major changes in open data norms must occur in glycoscience to make such data available to ML researchers. Additionally, many properties (e.g., functional, immunological, solvent-related, etc.) or multimodal datasets can be incorporated into future data and models to expand ML applications in this realm (Burkholz et al., 2021). Solvent-carbohydrate interactions, for example, remain poorly understood and theoretically important for understanding NMR data, but most public data remains in water (Klepach et al., 2015; Kirschner and Woods, 2001; Hassan et al., 2015). Future work should also explore more hierarchical structure graphs specifically tailored towards carbohydrates (Mohapatra et al., 2022). Overall, increasingly strong collaboration between glyco-focused scientists and ML-focused researchers is essential over the next decade in the field of glycoscience, as the quality and scope of structural and functional carbohydrate-specific databases must continue to improve and grow in parallel with the power of ML tools that utilize them.

## Broader Impact Statement

Improving understanding of carbohydrate structure and function relationships is critical to furthering advancements in fundamental sciences, biomedical applications, and climate change research, and ML can accelerate this progress. Our carbohydrate dataset is substantially larger and more diverse than previous carbohydrate NMR-structure datasets designed for ML applications. Thus, this dataset can be a useful tool for ML researchers seeking to build carbohydrate-specific models, as well as inspire closer collaboration between biochemistry-related researchers and ML researchers to improve the quality and scale of both datasets and models in this area. We do not foresee any possible negative consequences of our work.

## Acknowledgments and Disclosure of Funding

This work was supported by GlycoMIP, a National Science Foundation Materials Innovation Platform funded through Cooperative Agreement DMR-1933525.

## Data Availability

GlycoNMR is freely available for academic purposes. The full datasets are publicly available and can be downloaded at `https://github.com/Cyrus9721/GlycoNMR/`. The data will be permanently hosted on this GitHub repository under a CC BY 4.0 license, and we commit to answering inquiries about this dataset in a timely manner.

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

## Appendix A. Further Descriptive Analyses of Each Dataset

In this section, we provide a detailed data analysis of `GlycoNMR`, focusing on both the quantity and variety of monosaccharides within our dataset.

### A.1 Histogram distribution of carbohydrate lengths in both datasets

We further analyze the data volume of `GlycoNMR`. We plot the distributions of the number of monosaccharides that every carbohydrate contains in both GlycoNMR.Exp and GlycoNMR.Sim. In Figure 5, we use 'length of glycan' to denote the number of monosaccharides that the carbohydrate contains. We observe both histograms exhibit a right-skewed distribution in the length of the glycan. This indicates that `GlycoNMR.Exp` contains a greater proportion of small and middle-sized carbohydrates than large-sized carbohydrates. Therefore, existing MRL methods may be biased towards smaller carbohydrates. We further investigate the potential impact of representational bias related to carbohydrate size in Appendix J, where we validate the generalizability of a trained MRL to carbohydrates of its unseen sizes.

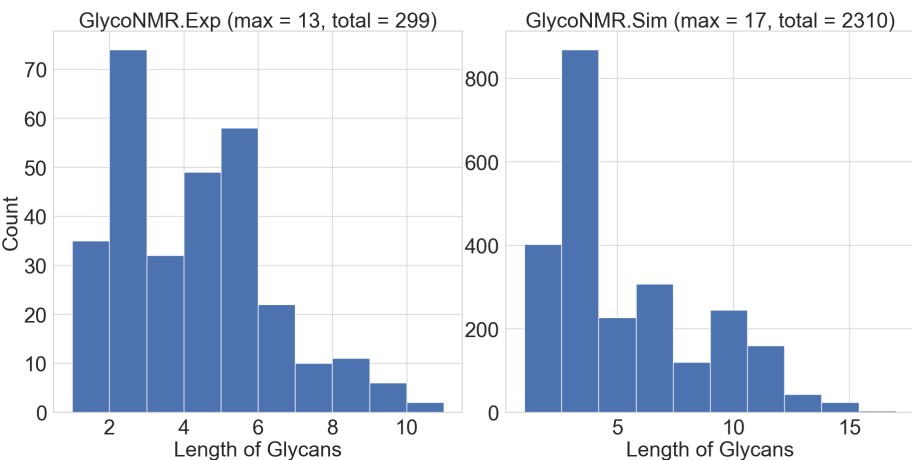

Figure 5: Distribution of glycan length in both datasets. The horizontal axis indicates the number of monosaccharides in the carbohydrate. The vertical axis indicates the corresponding number of carbohydrates presented in the dataset.

### A.2 Percentage of monosaccharide types in both datasets

We investigate the diversity of monosaccharide types in `GlycoNMR`. For each dataset, we count the occurrence of all monosaccharides and present the percentage of the top eight most frequently appearing monosaccharides in Table 5. The entry "Others" represents the category of relatively infrequently appeared monosaccharides, including stem type: ManA, Neu, GalN, Ara, etc. We demonstrate that `GlycoNMR` covers the most commonly occurring stems of monosaccharides as introduced in (Chaplin, 1986) for example.

Table 5:  Percentage of the most common monosaccharide unit types in the two datasets

| GlycoNMR.Sim | | GlycoNMR.Exp | |
|---|---|---|---|
| Monosaccharide | Percentage | Monosaccharide | Percentage |
| Glc | 18.86% | Gal | 19.73% |
| Gal | 17.5% | Glc | 17.7% |
| GlcNAc | 12.18% | GlcNAc | 12.21% |
| Fuc | 12.1% | Rha | 11.06% |
| Xyl | 8.51% | Man | 6.81% |
| Man | 6.23% | Fuc | 4.87% |
| GlcA | 6.19% | Kdo | 4.78% |
| GalA | 5.49% | GlcA | 4.42% |
| Others | 12.94% | Others | 18.42% |

## A.3 Feature statistics

In this section, we present detailed feature statistics for both *GlycoNMR.Exp* and *GlycoNMR.Sim*. Specifically, we show the percentage of atom-level features and monosaccharide-level features. For the atom level feature (first-row and the third-row of Figure 6), we present the proportional distribution of values for atom type, carbon atom position, and hydrogen atom position. For the description of atom identity (top left), 'other' indicates other types of atoms, including nitrogen, phosphorus, and sulfur. For the description of carbon atom position (top middle), 'Other' indicates the off-ring carbons. Similarly, for the description of the hydrogen atom position (top right), 'Other' indicates off-ring hydrogens. For the monosaccharide level feature (the second row and the fourth-row of Figure 6), we included Anomer (bottom left, indicates the hydroxyl group), Configuration (bottom middle, indicates Fischer project information), and Ring Size (bottom right, number of in-ring carbons) as introduced in Table 2. The 'N/A' of each pie chart indicates that the information is not contained in the PDB file.

## A.4 Ring Position vs. Shift Relationship

We investigate the relationship between the ring position and the NMR shift values. We plot the distribution of the NMR shift values by carbon and hydrogen ring positions. For both Figure 7 and Figure 8, the x-axis indicates the ring position of the atom (Carbon / Hydrogen), and the y-axis indicates the NMR shift values of the corresponding atoms. We notice that the distribution of NMR shift values for the ring positions C1 and C6 significantly vary from those of C2, C3, C4, and C5, similarly of H1 and H6 to H2, H3, H4, and H5. A fundamental factor that determines the NMR shift value is the atom's electronic environment, especially bonded or non-bonded interactions within 1-3 atom distances away from the atom of interest.

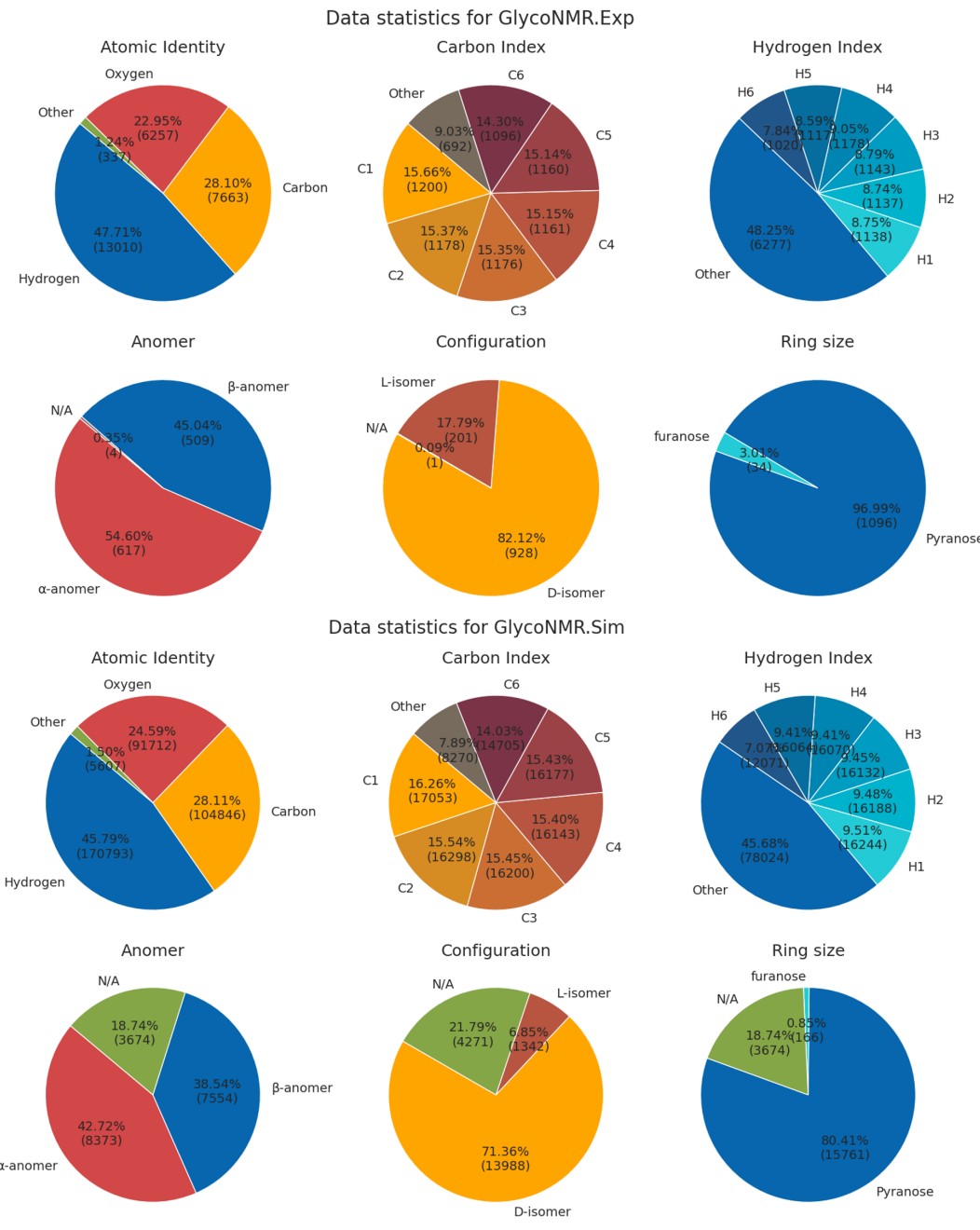

Figure 6: Data statistics for *GlycoNMR.Exp* and *GlycoNMR.Sim*.

## Appendix B. Details on Features Tables

In this section, we present a comprehensive description of the processed PDB file, including the curated features mentioned in Section 2 and Section 3.2. For each feature, we provide its data type along with a detailed explanation. Lines 1-8 in Table 6 record attributes presented in the original PDB file. We incorporate the Atom_name and Atom_type as components

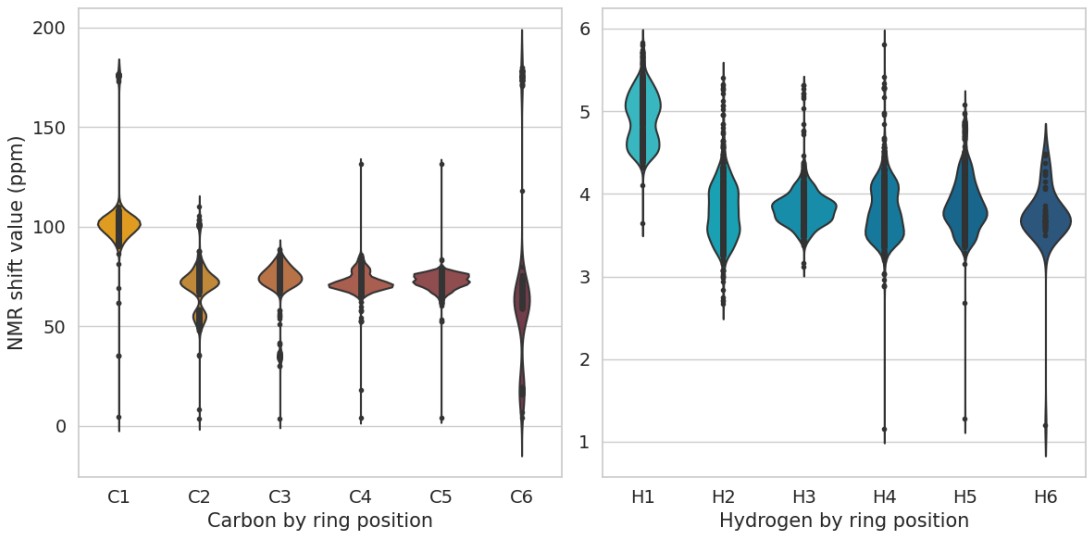

Figure 7: NMR shift value by ring position for *GlycoNMR.Exp*

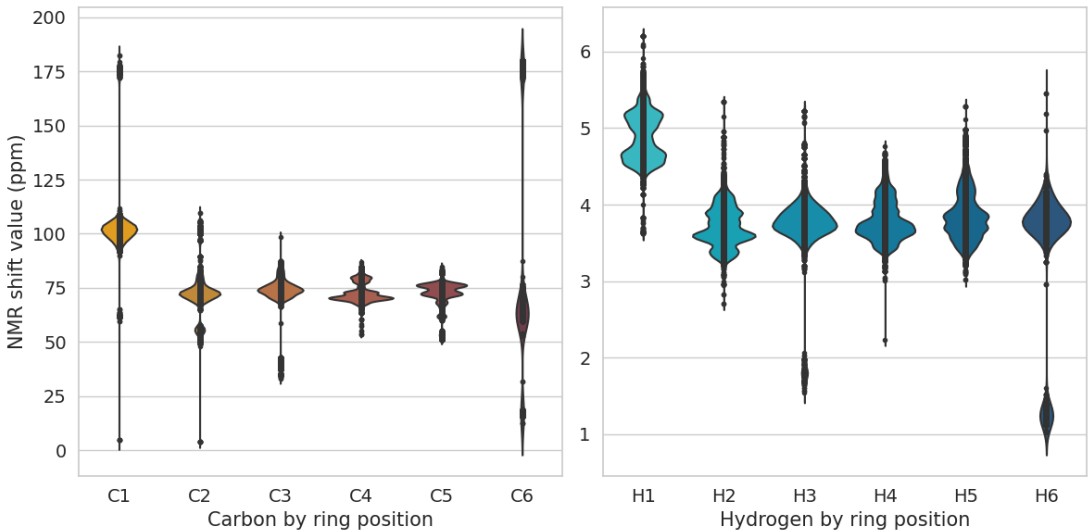

Figure 8: NMR shift value by ring position for *GlycoNMR.Sim*

of the node features. Coordinate x, y, and z is used as spatial information to construct the MRL models. Lines 9-15 record the processed node features as introduced Table 2. Lines 15-25 describe the feature: Modifications that are used in GlycoNMR.Sim. On curating the feature Modification, we first identify the modification group using Lineage, Atom_num, Residue_name, and atom connectivity. Then, we calculate each atom's distance(atom path) to the identified modification group, set up several distance thresholds to convert them into categorical values, and incorporate them as node features. Notice that the atom connectivity information is generally missing in GlycoNMR.Exp, thus it can be ambiguous to match the atoms to their corresponding modification groups, and we omitted this feature for now in the smaller Glycosciences.DB-sourced dataset only (in contrast, Modification was included in the

GODESS-sourced dataset). Future databases of new experimental results in carbohydrate NMR spectra should seek to improve the clarity in this area, such as with more uniform standards in data annotation by the original uploaders. Last, we use the labeled in-ring atoms' NMR shift as ground truth values.

Table 6: Detailed feature description

| Value | Datatype | Descriptions |
|---|---|---|
| Atom_num | Numerical | Atom index number in the carbohydrate |
| Atom_name | Categorical | Atom name that also indicates its within-monosaccharide position index |
| Residual_name | Categorical | Three letters abbreviation of monosaccharide name |
| Residual_num | Numerical | Monosaccharide order number assigned |
| x | Numerical | X coordinate of the atom |
| y | Numerical | Y coordinate of the atom |
| z | Numerical | Z coordinate of the atom |
| Atom_type | Categorical | Chemical element type of the atom |
| Residual_accurate_name | Categorical | Full name of monosaccharide or modification group that atom belongs to |
| Lineage | String | Lineage (linkage) information of the current residue |
| Ac_component | Categorical | Whether atom is in an Ac modification |
| bound_AB | Categorical | Anomeric orientation of hydroxyl group |
| fischer_projection_DL | Categorical | Fischer convention |
| reformulated_standard_mono | Categorical | Monosaccharide stem name |
| carbon_number_PF | Categorical | Number of ring carbons (ring size) |
| Me_min_atom_distance | Numerical | Distance of the shortest atom path to Me modification group |
| Me_min_atom_path | Categorical list | The shortest atom path to Me modification |
| Ser_atom_distance | Numerical | Distance of the shortest atom path to Ser modification group |
| Ser_atom_path | Categorical list | The shortest atom path to Ser modification |
| Ac_min_atom_distance | Numerical | Distance of the shortest atom path to Ac modification group |
| Ac_min_atom_path | Categorical list | The shortest atom path to Ac modification |
| S_min_atom_distance | Numerical | Distance of the shortest atom path to S-related modification group |
| S_min_atom_path | Categorical list | The shortest atom path to S-related modification |
| Gc_min_atom_distance | Numerical | Distance of the shortest atom path to Gc modification group |
| Gc_min_atom_path | Categorical list | The shortest atom path to Gc modification |
| main_ring_shift | Numerical | Chemical shift values of all labeled main ring atoms |
| shift | Numerical | Chemical shift values of all labeled atoms |

## Appendix C. Shapley analysis of Feature contributions

We calculate the Shapley values in Table 7 for the atomic-level and monosaccharide-level features as we introduced in Table 2, following the implementation method in (Štrumbelj and Kononenko, 2014). We noticed that, in general, all Shapley values are positive. Among all the features, the ring position of both carbon and hydrogen atoms plays a significant role in the NMR shift prediction. In addition, incorporating the stem type of the monosaccharides in the 2D GNN can moderately decrease the prediction error. The remaining features, such as modification group, anomer, configuration, and ring size, have a relatively minor impact on overall model performance.

Table 7: Shapley value of the carbohydrate-informed features for the 2D-based GNN models on GlycoNMR.Exp and GlycoNMR.Sim. Each column reports the Shapley values of the corresponding features.

| GlycoNMR.Exp | Ring position | Modification | Stem type | Anomer | Configuration | Ring size |
|---|---|---|---|---|---|---|
| $^1$H | 0.457 | N/A | 0.088 | 0.061 | 0.009 | 0.008 |
| $^{13}$C | 16.852 | N/A | 2.640 | 0.515 | 0.257 | 0.085 |

| GlycoNMR.Sim | Ring position | Modification | Stem type | Anomer | Configuration | Ring size |
|---|---|---|---|---|---|---|
| $^1$H | 0.387 | 0.014 | 0.112 | 0.187 | 0.051 | 0.014 |
| $^{13}$C | 13.007 | 0.321 | 3.619 | 0.465 | 0.199 | 0.055 |

## Appendix D. Possible Future Research Topics

In this section, we provide several unexplored glycoscience-related research topics that `GlycoNMR` can be used for. We believe these topics can potentially benefit the overall ML and glycoscience community.

**Overview**   A common problem in glycosciences is matching structure to NMR spectra. For example, a scientist may want to verify they have generated the correct structure in the laboratory, by examining a compound's spectra after synthesis. NMR spectral peak positions provide key features for carbohydrate structure identification, including the stereochemistry of monosaccharides, glycosidic linkage types, atomic interactions and couplings, and conformational preferences. Individual atoms (with net spin) in a carbohydrate generate the key spectral peaks for structure interpretation, which in practice in carbohydrates is typically the central ring carbon and hydrogen atoms, plus certain modification groups. Chemical shift values reported in ppm units are also independent of spectrometer frequency and thus comparable across labs and equipment settings. In carbohydrates, usually, only the hydrogen 1H and carbon 13C nuclei shifts are measurable, making spectra harder to interpret than protein spectra where nitrogen and phosphorus shifts are also accessible (Toukach and Ananikov, 2013). As another challenge specific to carbohydrates, carbohydrate NMR peaks are constrained to a much narrower region of spectra range than proteins, making them harder to separate and leading to an over-reliance on manual interpretation (Toukach and Ananikov, 2013). The development of theoretical and computational ML-based tools that can utilize large datasets to find and predict relationships between carbohydrate structure and its NMR parameters is a high priority for the field.

**Customized models for carbohydrate data**   Models specifically designed to accommodate the unique characteristics and structure of the carbohydrate data are important to develop. As introduced in Section 2, carbohydrates are a special type of biomolecule that is formed via the condensation reactions of monosaccharides. We conduct heavy feature engineering to extract the monosaccharide-related features, and our experimental results in Table 3 have already demonstrated the usefulness of monosaccharide information (stem type) in NMR shift prediction. However, we incorporate them as atom-level features in our baseline and the 3D-based MRL models. In this case, the existing models may fail

to capture the spatial information between monosaccharides, and more neural network layers corresponding to the structural hierarchies inherent to carbohydrates could improve prediction quality in future work. On the other hand, a carbohydrate's unique atoms-to-monosaccharides-to-carbohydrate characteristic inherently satisfies a hierarchical graph structure, so the information is partly captured in the current implementation. We believe that developing a customized MRL model (e.g., learning representations for both atoms and monosaccharides) can help learn a better node representation for accurate NMR shift predictions in future work.

Theoretically advancing NMR-based structural analysis approaches in ML directions requires having a comprehensive database where the same base monosaccharide units have various neighboring units or modification groups swapped out or removed across data entries, in order to see how the spectra changes as various components are combined or removed to better train models. Such comprehensive databases have been established and well-studied in protein ML research, but a lack of ML-friendly databases and poor open access data norms have hindered parallel progress in carbohydrates. While our database is certainly not comprehensive and complete, with carbohydrates being more diverse and varied than any other class of biomolecule, our approximately 2600 NMR spectra and structure files tailored for ease of use in ML pipeline is the first of its size for ML studies.

For additional ideas for boosting the data size and quality in future work: by our assessment, GODESS provides the best balance of accuracy, efficiency, and accessibility for the simulation of 1D NMR of carbohydrates. However, as with any simulation method, it likely has some biases and simplifications not seen in experimental data which are difficult to reveal without a large experimental dataset for comparison. Thus, it is important for future work to expand this dataset to include simulation datasets from other sources (e.g. CASPER (Furevi et al., 2022)), as well as to expand the experimental dataset for comparison to the theoretical predictions. The experimental dataset expansion will necessitate a serious and concentrated effort on the part of glycoscience researchers to improve the open data norms of their field.

**Predicting NMR spectra:** As presented in Section 3.1, extensive data annotation is required for preparing the atom-level carbohydrate NMR chemical shift data. Notably, for annotating each carbohydrate, the key step is to match the monosaccharides present in the PDB (Protein Data Bank) structure file to the monosaccharides present in the NMR (Nuclear Magnetic Resonance) chemical shift file. This step not only demands significant effort but also necessitates domain expertise, but will continue to do so at least until the experimental glycoscience field adopts more uniform standards in data files.

In the field of glycosciences, the ideal scenario is to predict the full continuous spectrum (peak widths and noise included) depicted in Figure 2 (b) and (c) directly from the carbohydrate structure. In our case, the NMR chemical shift prediction problem of just peaks is reformulated as graph-regression tasks with promising initial performance. The biggest improvements in this direction will necessitate both increasingly larger and more diverse experimental datasets, as well as model innovations.

## Appendix E. Model Setup and Computation Resources

All data processing and model training is performed on a Linux workstation with an Intel Core i7 CPU, 32GB memory, and two GeForce RTX 3090 GPUs. Our entire training time for all models in aggregate was on the scale of several hours. We also provided a detailed run-time information per epochs in Table 12.

## Appendix F. Disclaimer on GlycoNMR licensing

**Disclaimer on GlycoNMR.Exp**  GlycoNMR.Exp is freely available under CC BY 4.0 licensing and can be downloaded within this link. GlycoNMR.Exp is laboriously curated from Glycoscience.DB to facilitate machine learning research on NMR shift predictions of carbohydrates. Glycosciences.DB experimental data uploaded from various labs can be downloaded within this link. Glycosciences.DB (Böhm et al., 2019), as part of the Glycosciences.de (Toukach et al., 2007) portal, is provided for the glycoscience community with unrestricted open access intent. According to (Toukach et al., 2007): "All glycan-related scientific data of the GLYCOSCIENCES.de portal are freely accessible via the Internet following the open access philosophy: 'free availability and unrestricted use'."

**Disclaimer on GlycoNMR.Sim**  GlycoNMR.Sim is freely available under CC BY 4.0 license and can be downloaded within this link. GlycoNMR.Sim is extensively curated from the simulation software GODESS. The GODESS experimentally-informed simulation data without preprocessing can be downloaded within this link. GODESS simulation output is free to use and does not have a license (see https://glic.glycoinfo.org/software/), if proper attribution to the references is done.

## Appendix G. RMSE formula for Benchmarks

The RMSE was calculated according to the usual equation in all results presented throughout the manuscript:

$$RMSE = \sqrt{\sum_{i=1}^{N} \frac{(y_i - \hat{y}_i)^2}{N}}.$$

Where $y_i$ is the recorded NMR chemical shift, $\hat{y}_i$ is the prediction from our GNN model on the $i^{th}$ atom from the test set, and N is the number of the test data points.

## Appendix H. Evaluation Using MAPE

We also assessed the NMR shift prediction performance using the Mean Absolute Percentage Error (MAPE), offering an alternative perspective to the RMSE metric. MAPE measures the percentage error and is defined as:

$$\text{MAPE} = \frac{100\%}{N} \sum_{i=1}^{N} \left| \frac{y_i - \hat{y}_i}{y_i} \right|$$

where $N$ is the number of the test atoms, and $y_i$ and $\hat{y}_i$ are the ground-truth and prediction of the $i^{th}$ test atom, respectively. On the **GlycoNMR.Sim** dataset, all 3D GNN models were run with their default hyper-parameters with the batch size set to 4, the number of layers set to 4, and the hidden channel size set to 128, and the cut-off distance set to 4.0. On the **GlycoNMR.Exp** dataset, all 3D GNN models were run with their default hyper-parameters with the batch size set to 4, the number of layers set to 2, the hidden channel size set to 64, and the cut-off distance set to 4.0. We observe that the overall MAPE of $^{13}C$ shift is lower than the overall MAPE of $^1H$ shift. This indicates that the 3D Graph Neural Network (GNN) models are relatively better at predicting carbon shifts than hydrogen shifts.

Table 8: NMR shift prediction performance evaluated using MAPE

| | GlycoNMR.Sim | | GlycoNMR.Exp | |
| --- | --- | --- | --- | --- |
| | $^{13}$C | $^1$H | $^{13}$C | $^1$H |
| ComENet (Wang et al., 2022) + *extra feat.* | 0.910±0.021 | 1.427±0.013 | 2.247±0.042 | 2.613±0.140 |
| DimeNet++ (Gasteiger et al., 2020a) + *extra feat.* | 0.873±0.019 | 1.413±0.077 | 1.829±0.087 | 2.575±0.182 |
| SchNet (Schütt et al., 2017) + *extra feat.* | 0.853±0.010 | 1.651±0.023 | 1.816±0.018 | 2.201±0.016 |
| SphereNet (Liu et al., 2022) + *extra feat.* | 0.820±0.013 | 1.438±0.036 | 1.925±0.069 | 2.745±0.123 |
| CASCADE (Guan et al., 2021) + *extra feat.* | 0.851±0.009 | 1.638±0.017 | 1.831±0.026 | 2.172±0.031 |

## Appendix I. Assessing the Quality of GODESS Simulation

**GODESS quality assessment on GlycoNMR.Exp**  This section showcases the NMR simulation quality of the GODESS software, which we used to construct the simulation dataset GlycoNMR.Sim. We validated the simulation results against the real-world experimental database Glycosciences.DB, from which GlycoNMR.Exp is curated from. In detail, we identified 59 simulated GlycoNMR.Sim carbohydrates (with an average carbohydrate length of 5.6) were also present among the entries of GlycoNMR.Exp experimental data. We compared the simulated NMR shifts with the experimentally measured NMR shifts. We present the comparison results in Figure 9. The RMSE between GODESS and experimental chemical shifts are 1.69 and 0.11 for $^{13}$C and $^1$H, respectively. This result indicates the relatively high quality of GODESS performance, with low simulation error between the predicted GODESS shifts and those carbohydrates observed in experimental NMR data. These findings validate the potential of GODESS as a tool in simulating NMR shifts on carbohydrates to produce training data for developing machine learning methods.

We note that the above assessment is limited by the small number of directly matched entries between the carbohydrates in GlycoNMR.Exp and GlycoNMR.Sim. Although this evaluation only uses 2.5 percent of carbohydrates in GlycoNMR.Sim, it provides a reassuring starting point. In the future, as more annotated experimental data become available, a more comprehensive validation should be conducted to evaluate GODESS across a broader range of carbohydrates.

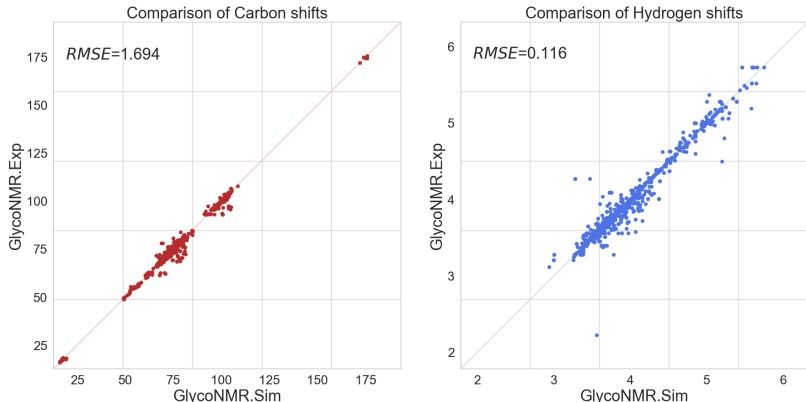

Figure 9: Agreement between chemical shift values for a subset of carbohydrates (N=59) that are present in both *GlycoNMR.Sim* (produced by the simulation software GODESS) and the experimentally-sourced dataset *GlycoNMR.Exp*. The x-axis indicates the chemical shift from GODESS and the y-axis indicates the experimental chemical shift value, thus comparing atoms that were matched and compared between each molecule that was both simulated and experimentally measured.

**GODESS quality assessment in existing literature** GODESS accuracy (RMSE) has been reported as 0.03 to 0.12 ppm for [1]H shifts for a set of test molecules (Kapaev and Toukach, 2015). GODESS documentation estimates moderate prediction quality of [13]C shift as approximately less than or equal to 1-2 ppm error dependent on the complexity and internal data quality of a given glycan. In the words of the creators of GODESS (Kapaev and Toukach, 2015): "[GODESS] has perceptible performance superiority over quantum-mechanical methods and provides better accuracy similar to that of other glyco-tuned empirical approaches." Generally, our performance assessment agrees with this previously reported GODESS validations on experimental data.

## Appendix J. Assessing the Generalizability of GlycoNMR.Exp-trained Model to Molecules of Previously Unseen Sizes

In both datasets longer carbohydrates are less abundant than smaller ones, see Figure 5. To investigate potential bias that may be caused in model training, we performed three experiments to investigate the generalizability of a trained 2D GNN to carbohydrates of previously unseen sizes. In each experiment, we split the carbohydrates into the training and test sets using a 4:1 ratio. In the first experiment, the carbohydrates in the training set are smaller (in terms of monosaccharide number) than those in the test set (mean carbohydrate length is 3.32 residues in training vs 7.47 in test). In the second experiment, the training set contained larger carbohydrates (mean carbohydrate length is 4.84 residues in training vs 1.45 in test). In the third experiment, the split was done randomly. The results are summarized in Table 9. It is observed that the test errors are noticable larger than the training error under asymmetrical training/test length conditions. However, the test errors are still the

same order of magnitude comparing to the training errors. Random split, allowing models to see more diverse sizes of carbohydrates, helps alleviate the problem.

Table 9: Assessing the effects of the length bias in GlycoNMR.Exp

| GlycoNMR.Sim | Ave. train/test carbohydrate length: 3.32/7.47 | Ave. train/test carbohydrate length: 4.84/1.45 | Random Split |
|---|---|---|---|
| Train RMSE $^{13}$C | 2.699 | 2.918 | 3.376 |
| Test RMSE $^{13}$C | 4.139 | 4.619 | 3.682 |
| Train RMSE $^{1}$H | 0.197 | 0.211 | 0.208 |
| Test RMSE $^{1}$H | 0.273 | 0.232 | 0.199 |

## Appendix K. Random Forest Baseline

Table 10: NMR chemical shift prediction benchmark using a random forest model (in RMSE). The code is provided on the Github repository.

| | GlycoNMR.Sim | | GlycoNMR.Exp | |
|---|---|---|---|---|
| | $^{13}$C | $^{1}$H | $^{13}$C | $^{1}$H |
| Random Forest | 2.446 | 0.132 | 4.117 | 0.178 |

We conducted a traditional ML baseline experiment using random forest to predict atomic NMR shifts. The features of each atom (represented as a node in its carbohydrate graph) follow the same initializing method as used for training the 2D GNN model. In addition, we follow the same splitting method as we did in Section 3.4. In general, the baseline model slightly underperforms relative to the 2D GNN model. This demonstrates the effectivenss of our feature engineering step in Section 3.2.

## Appendix L. Benchmark on Multi-task NMR Shift Prediction

We trained 3D GNN models to perform multi-task learning on both GlycoNMR.Sim and GlycoNMR.Exp. Each 3D-based model was trained to predict both $^{13}$C and $^{1}$H shifts. The results are summarized in Table 11. Compared to the single-task prediction results, the performance of all models drops noticeable.

Table 11: NMR chemical shift prediction benchmark using 3D MRL methods (in RMSE).

| | GlycoNMR.Sim | | GlycoNMR.Exp | |
|---|---|---|---|---|
| | $^{13}$C | $^{1}$H | $^{13}$C | $^{1}$H |
| ComENet (Wang et al., 2022) | 1.987 | **0.157** | **3.006** | 0.411 |
| DimeNet++ (Gasteiger et al., 2020a) | 1.954 | 0.199 | 3.696 | **0.185** |
| SchNet (Schütt et al., 2017) | **1.523** | 0.590 | 3.187 | 0.946 |
| SphereNet (Liu et al., 2022) | 2.258 | 0.169 | 3.364 | 0.638 |

## Appendix M. Running time comparison

Table 12: Running time(s) comparisons for 3D GNNs

| Dataset | ComeNet | DimeNet++ | SchNet | SphereNet |
|---|---|---|---|---|
| GlycoNMR.Sim | 7.564 | 20.581 | 3.615 | 31.831 |
| GlycoNMR.Exp | 1.257 | 2.312 | 0.754 | 2.032 |

The running time comparison of 3D GNN models, as well as the duration in seconds for each training epoch, is reported. For a fair comparison across the 3D-based GNN models in GlycoNMR.Sim dataset, we set the batch size to 4, the number of hidden channels to 128, and the number of layers to 4, in GlycoNMR.Exp, we set the batch size to 2, the number of hidden channels to 64, and the number of layers to 2.

## Appendix N. Data annotation supplements

In this section, we provide two supplemental repositories to help illustrate our data preprocessing pipeline. One of our major contributions is to extensively curate the raw files from the Glycosciences.DB- and GODESS-sourced datasets to make the GlycoNMR dataset friendly to machine learning researchers. To achieve this, we have made significant efforts in data preprocessing and provided a reproducible protocol for use in curating future carbohydrate-related NMR/structure databases.

### N.1 Overview

We summarize the data preprocessing pipelines on Glycosciences.DB as the following five steps:

1. We manually and semi-automatically checked the carbohydrate data scraped from Glycosciences.DB, and kept carbohydrates with complete or NMR peak shift lists.We selected molecules from Glycosciences that met specific criteria to ensure data quality. Specifically, we included molecules with no more than two missing atomic shift entries per residue and excluded those with gross or obvious errors in shift reporting, such as nonsensically large values or heavily duplicated shifts. Additionally, we required both $^{13}$C and $^{1}$H shifts to be reported, along with an accompanying structure file. Out of the 3,400 entries considered, the data quality tended to be polarized: entries either had many missing atomic shifts or very few, with minimal cases of intermediate data quality. Consequently, our selection criteria were not an arbitrary cut-off but rather a clear distinction based on the observed data distribution.

2. We consolidated all the PDB files as well as the NMR label files into a uniform machine interpretable format. The original data files are in various formats as they came from diverse sources.

3. We examined the carbohydrates with branched monosaccharide chains, and manually matched the monosaccharide IDs from the PDB file and the NMR label file.

4. We trained a simple 2D GNN model and predicted the NMR chemical shifts for each annotated atom.

5. We examined the carbohydrates with the highest ranked errors and manually checked each one for potential mismatch problems. Then, we fixed the identified mismatch problems based on the domain knowledge. This was repeated many times until we were not able to identify any mismatch problems. To fix some monosaccharide mismatch problems, we needed to go back to steps 2/3/4.

The data preprocessing pipeline in GODESS is relatively similar to the Glycoscience.DB. We constructed a more streamlined semi-automatic pipeline to annotate the GODESS dataset since the dataset is generated from a single simulation software with more consistent formatting. We introduced this pipeline in our released repository provided below.

To further demonstrate our efforts, we released two repositories for reference on data cleaning, processing, and annotating:

Creating GlycoNMR.Sim from the GODESS (`https://github.com/Cyrus9721/GODESS_preprocess`)

Creating GlycoNMR.Exp from the Glycosciences.DB (`https://github.com/Cyrus9721/GlycoscienceDB_preprocess`).

The data preprocessing steps are provided in detail in the README.md file.

## N.2 An annotation example from GlycoNMR.Exp

For carbohydrate file DB26380, we need to manually annotate the PDB file by assigning each central ring carbon and hydrogen atoms with their corresponding shift values, which is stored in the NMR label file. To achieve this, we need to associate the atoms' parent monosaccharide IDs between the two files. We first draw a sketch of the carbohydrate structure consisting of the basic monosaccharide components from the CSV file using the linkage information. Atoms with the same linkages are from the same monosaccharides. For example, atoms from lines 13-19 belong to monosaccharide B-D-GLCPN. We utilize linkage information to identify monosaccharide components but not monosaccharide names such as 'B-D-GLCPN' because, in some scenarios, the same monosaccharide name may indicate different monosaccharide components (i.e., there can be multiple monosaccharide units with the same name in a carbohydrate, but the linkage information can be used to tell them apart for NMR shift matching purposes). For example, lines 62-67, 68-73, and 74-79 of the NMR label file refer to three separate monosaccharide unit components, that are parents of different sets of atoms and appear in different locations of the carbohydrate chain, but still have the same monosaccharide chemical name. DB26380's sketch plot can be found on the 8th page (plot number 23) of our annotation document for branched carbohydrates. Second, we again inspect the PDB file and match the monosaccharide components with the help of the SWECON information which provides additional secondary linkage information at the bottom of Glycosciences.DB PDB file (lines 306-315) and our domain expertise. Then, for another example of a common issue causing mismatches between monosaccharide shift and structure, in DB26380, we noticed that the Phosphoryl group 'PO3' (lines 39-42, 64-67) is treated as a monosaccharide component in the PDB file despite not being a monosaccharide, therefore the monosaccharide shift file ID ordering 3 and 13 should be disregarded when

comparing to the PDB structure file, and the 4th monosaccharide residue in the PDB file should instead be matched with the 3rd monosaccharide parent and its atom components in the NMR label file. A detailed match is presented in our PDF document mentioned above. Then last, when all parent monosaccharides are correctly matched between structure and shift files, we assign the corresponding monosaccharide atoms' shift from the label file to the PDB file by their atom names.

## N.3 An annotation example from GlycoNMR.Sim

For glycan: 'aDXylp(1-6)bDGlcp(1-4)[aLFucp(1-2)bDGalp(1-2)aDXylp(1-6)]bDGlcp(1-4)[aL Fucp(1-2)bDGalp(1-2)aDXylp(1-6)]bDGlcp(1-4)xDGlca' and its corresponding PDB file, monosaccharide bond linkage '(1-4)' indicates the carbon with position number 1 is connected to the carbon with position number 4 via a dehydration synthesis reaction, where 'xDGlca' is the precursor monosaccharides (in other words 'root'). From line 223 of the PDB file, we notice that atom 1 is connected to atoms 28 and 2; this indicates that the monosaccharide with ID 2 is connected to a monosaccharide with ID in the following bounds (C1 - O4 - C4), where C indicates the carbon and O indicate the oxygen and the following number indicates the ring position. In this case, from the 3rd line of the label file, we can match the monosaccharides residue 'b-D-Glcp' from the label file to the monosaccharides ID 2 in the PDB file using the linkage information ', 4' which indicates the following bounds (C1 - O4 - C4). Then, again, we assign the corresponding monosaccharide atom's shift from the NMR label file to the PDB file by its atom name.

## N.4 Additional details on annotating GlycoNMR.Exp

1. We originally gathered 301 carbohydrates with complete structure files from Glycosciences.DB. Two carbohydrates were dropped. One was dropped because of the unconventional naming ways of atoms, while another was because of the rare, single occurrence of certain monosaccharide types in the dataset that created sampling issues. We then use the remaining 299 carbohydrates for the following steps.
2. We split the files by linear carbohydrates and branched carbohydrates by inspecting their formula. Out of 299 carbohydrates, we have 184 linear glycans and 115 branched carbohydrates. Both types of carbohydrates had issues with different residue orderings between the NMR file and the PDB structure file. We focused on solving this problem first and manually double-checked to make sure that we corrected all the ordering mismatches.
3. For branched glycans, we also manually inspected every pair of NMR and PDB files, and we identified 40 glycans with further mismatching issues (mismatches at the atom level and mismatches of the connections of Ac groups to specific atoms within monosaccharides). This mainly happens because of the different naming criteria used in different labs for atoms' and monosaccharides' name abbreviations describing atoms or residues that are identical.
4. We again inspected every pair of NMR and PDB files for linear glycans and identified 39 cases containing mismatch problems (at the atom level and at the connections of Ac groups to specific atoms within monosaccharides). The above efforts require us to go through each entry manually and perform detailed quality checks on all experimental data files and a large set of simulation data files, in tandem with point-by-point investigation of top ranked outliers in the shift prediction plots across iterative model runs to make sure prediction error

was not due to annotation problems. Hence, we believe that the chance of mismatch in the annotated dataset is minimal, if not impossible. Before resolving these annotation issues, the model RMSE was also an order of magnitude worse for both C and H chemical shifts. A digitalized example can be found here.

## Appendix O. Example codes and demos

We provide four Jupyter Notebook demos in the GitHub repo for detailed instructions. They introduce step by step on how to utilize the GlycoNMR.Sim and GlycoNMR.Exp datasets to train a 3D or 2D GNN model.

Train a 2D-based GNN model on GlycoNMR.Sim: `https://github.com/Cyrus9721/` `GlycoNMR/blob/main/2D_example_Sim_GlycoNMR.ipynb`.

Train a 2D-based GNN model on GlycoNMR.Exp: `https://github.com/Cyrus9721/` `GlycoNMR/blob/main/2D_example_Exp_GlycoNMR.ipynb`.

Train a 3D-based GNN model on GlycoNMR.Sim: `https://github.com/Cyrus9721/` `GlycoNMR/blob/main/3D_example_Sim_GlycoNMR.ipynb`.

Train a 3D-based GNN model on GlycoNMR.Exp: `https://github.com/Cyrus9721/` `GlycoNMR/blob/main/3D_example_Exp_GlycoNMR.ipynb`.

