# OpenReview forum: "GlycoNMR: Dataset and Benchmark of Carbohydrate-Specific NMR Chemical Shift for Machine Learning Research"
_DMLR — Accepted by DMLR_

### Review · Reviewer_F75f · 2024-05-12

**Recommendation:** 4
**Confidence:** 3

**Summary Of Contributions:**

This paper proposes GlycoNMR, which is a chemical shift dataset with a state-of-the-art size. To be more accurate the main contributions are three-fold

1. The above-mentioned dataset, with a state-of-the-art size and friendly to ML algorithms.

2. A set of features with experimentally verified performance improvement for graph-based models.

3. A set of benchmarks for the liturate machine learning algorithms.

**Strengths:**

Please see the above block for the strengths.

**Audience:**

Yes

**Broader Impact Concerns:**

No concern within my scope.

**Claims And Evidence:**

Yes!

**Datasets And Benchmarks:**

Yes to all questions above.

**Extended Submissions:**

I haven't see any publication that this paper is an extension of it.

**Limitations:**

From an ML researcher's point of view:

  1. It is not clear enough to the researchers without solid chemistry background to trade-off between the two datasets (.Exp and .Sim). I think the keep problem is that ML researchers have no idea about how accurate DFT or GODESS is (even though this is common knowledge for most computational chemistry researchers). Thus, I request the authors to add a section to describe how accurate are the GODESS algorithms and provide a guidance on how to trade-off between the two datasets. The best case would be showing some RMSE data for GODESS or DFT.

  2. For the ablation study of this paper, why does the RMSE for the two datasets have significant differences in C13 chemistry shift (~1.4 vs ~2.6)? For the simulated dataset, I saw at least two possibilities: (1) The data distribution is not the same for the two datasets.  (2) the MRL algorithm actually is trying to fit the GODESS algorithm instead of the real chemical shift data. Please clarify this in the revision.


From a chemistry researcher's point of view

  1. It is not clear to me how the .exp dataset identifies the chemical shift of each atom and how the atoms are numbered. When doing NMR experiments, I will get a plot that includes all peaks.

  2. It is not clear how to generate data (the model input of a molecule) for prediction tasks. I suggest the author to also add a guideline to make the ML models using this dataset more friendly to the users.

**Requested Changes:**

Please see the block below for the questions/ limitations and requested changes.

**Strengths And Weaknesses:**

Strength:

  1. This paper addressed an important problem for machine learning research in NMR: the lack of ML-friendly datasets to train/ eval the models. Especially, the lack of large datasets usually blocks ML researchers from developing larger models owing to overfitting.

  2. The Benchmarks in the paper proved a better insights of ML algorithms

Weakness:

  1. The presentation needs improvements to be friendly to both ML and chemistry communities.

---

### Review · Reviewer_bVVi · 2024-05-27

**Recommendation:** 2
**Confidence:** 3

**Summary Of Contributions:**

The authors curate and annotate two new datasets (GlycoNMR.Exp and GlycoNMR.Sim) for training machine learning models to predict nuclear magnetic resonance (NMR) chemical shifts of carbohydrates. The first dataset (GlycoNMR.Exp) is selected from an existing dataset Glycosciences.DB. The second dataset (GlycoNMR.Sim) is constructed via simulation using an existing method (GODESS).

The datasets are annotated by matching monosaccharides in each carbohydrate’s structural profile in Protein Data Bank (PDB) and the corresponding NMR shift data, which utilizes the domain knowledge and manual efforts of the authors. In addition, the authors design a set of chemically informed features for NMR shift prediction, perform benchmark evaluations for several 2D and 3D graph neural networks on the datasets, and illustrate the importance of different chemical features on the prediction performance.

**Strengths:**

Please see the “Strengths And Weaknesses” section above.

**Audience:**

Yes

**Broader Impact Concerns:**

I do not find any concerns over the broader impact of this work.

**Claims And Evidence:**

The quality statement of the dataset is not robustly justified. The authors should discuss and potentially use quantitative metrics to measure the quality of the curated dataset (e.g., dataset distribution bias, accuracy of simulated NMR shifts, etc.).

The authors should also fix the broken links of the GitHub repo and make sure that other researchers can rigorously reproduce their benchmark evaluation results.

**Datasets And Benchmarks:**

The authors provide the GitHub repo of their dataset and code to train graph neutral networks (link: https://github.com/Cyrus9721/GlycoNMR). However, as of May 27, 2024 the links in the repo to download the annotated datasets are broken and the links to view the two preprocessing GitHub repos are also broken. As a result, I cannot run the authors’ code and reproduce their results.

**Extended Submissions:**

This submission is not an extension of a previously published work.

**Limitations:**

Please see the “Strengths And Weaknesses” section above.

**Requested Changes:**

Please see the “Strengths And Weaknesses” section above. Please find my additional comments below.

1. Please explain what “nearly complete” NMR peak shift means (Section L.1) and discuss whether other data selection criteria are applied.

2. Although I thank the authors for sharing their handwritten notes of the annotation process, it’d be clearer to the readers if the authors can organize their annotation and matching process in a digital PDF with computer generated graphs instead of handwritten notes. Providing the digital PDF for a few examples is sufficient.

3. Please provide sufficient details on training the graph neural networks to allow other researchers to rigorously reproduce the evaluation results reported in this paper.

**Strengths And Weaknesses:**

Strengths

1. The curated dataset is the first of its kind and is designed for training machine learning models to predict NMR shifts of carbohydrates.

2. The paper is clearly written and easy to understand for non-specialists in the field of molecular machine learning and glycoscience.

3. The authors demonstrate considerable domain expertise in glycoscience and due diligence in their annotation process.


Weaknesses

1. As of May 27, 2024, the links in the paper’s GitHub repo to download the annotated datasets are broken and the links to view the two preprocessing GitHub repos are also broken. As a result, I cannot run the authors’ code or reproduce their benchmark evaluation results.

2. Although the authors’ diligent effort is appreciated, the dataset and paper may be of limited interest to readers of this journal. The problem domain (NMR shift prediction for carbohydrates) is too specialized and does not seem to enable transfer learning to related fields. The data modality (chemical structure) is monolithic and is unlikely to be used by ML researchers outside this field as a benchmark dataset. The annotation methodology, although clearly described, does not provide many generalizable insights to other fields or applications.

3. The size of the curated dataset is small, and the quality of the NMR shift data is not assessed. Only 299 carbohydrates have experimental NMR shift data as ground truth labels, which is much smaller than the size of the source dataset with 3,400 carbohydrates, as the authors point out. The NMR shift data of the other 2,310 carbohydrates are generated from the simulation method GODESS, which is less reliable and less accurate than both experimental data and the traditional density functional theory (DFT) computation results. The authors also do not evaluate the accuracy of the simulated NMR shift data against the ground truth experiment data. There is a concern that the limited accuracy of the simulated NMR shift data may be the bottleneck for training accurate and generalizable machine learning models.

4. On a related note, the authors do not address or provide quantitative metrics to measure the quality of their annotation process, e.g., how much mismatch (between the NMR and PDB chemical structure file) is removed by the annotation process, and how much mismatch (or uncertain match) remains in the annotated dataset.

5. The authors should discuss whether their data curation process introduced any bias in the dataset due to missing data issues or data selection criterion. For instance, the authors reported that only 299 out of the 3400 carbohydrates from the source dataset Glycosciences.DB have both chemical structures and complete 1-H and 13-C shifts and are included in the curated dataset. Does this significant data missingness issue lead to a biased representation of the carbohydrates in any dimension (e.g., smaller molecules are over-represented than larger ones) and affect the observations in the benchmark evaluations? The authors should perform a dataset audit and analyze the effects of any potential biases in detail.

6. Though the authors are familiar with graph neural networks, many other important methods (both machine learning models and computational chemistry methods such as density functional theory) are left out in the literature review and benchmark evaluations. The authors should include other ML methods such as ShiftML (Paruzzo et al., 2018), IMPRESSION (Gerrard et al., 2020) in the benchmark evaluations. More relevant ML models can be found in a recent review paper by Cortés et al. (2023). These methods and evaluations should be included to rigorously show that the curated datasets are of general interests for a wide group of researchers.

7. The authors only used RMSE as the metrics for benchmark evaluations, which is largely insufficient for a systematic evaluation. To compare the prediction accuracy of different classes of molecules, the authors are recommended to report a relative percentage error (such as MAPE), which accounts for the different baseline values of NMR shifts.

---

### Review · Reviewer_sUdT · 2024-05-27

**Recommendation:** 3
**Confidence:** 2

**Summary Of Contributions:**

This paper introduces a  carbohydrate NMR dataset named GlycoNMR and design a set of chemically-informed features for carbohydrates. The paper also benchmarks multiple 3D-based MRL methods on GlycoNMR.

**Strengths:**

This paper are well-written with detailed description on the development of the dataset and benchmarks of the proposed dataset. The dataset page is well illustrated with good documentation.

**Audience:**

Yes

**Claims And Evidence:**

Yes

**Datasets And Benchmarks:**

The includes a documentation and URL for reviewer access. But the paper does not include a a hosting, licensing and maintenance plan.

**Extended Submissions:**

N/A

**Limitations:**

The paper does not include a a hosting, licensing and maintenance plan.

**Requested Changes:**

The paper need to include a a hosting, licensing and maintenance plan.

**Strengths And Weaknesses:**

This paper are well-written with detailed description on the development of the dataset and benchmarks of the proposed dataset. The dataset page is well illustrated with good documentation.

But the paper does not include a a hosting, licensing and maintenance plan.